# SCONE: A Food Scooping Robot Learning Framework with Active Perception

**Yen-Ling Tai**
National Yang Ming Chiao Tung University
yling.cs10@nycu.edu.tw

**Yu Chien Chiu**
National Yang Ming Chiao Tung University
ychiu2@cougarnet.uh.edu

**Yu-Wei Chao**
NVIDIA
ychao@nvidia.com

**Yi-Ting Chen**[*]
National Yang Ming Chiao Tung University
ychen@cs.nycu.edu.tw

**Abstract:**
Effectively scooping food items poses a substantial challenge for current robotic systems, due to the intricate states and diverse physical properties of food. To address this challenge, we believe in the importance of encoding food items into meaningful representations for effective food scooping. However, the distinctive properties of food items, including deformability, fragility, fluidity, or granularity, pose significant challenges for existing representations. In this paper, we investigate the potential of active perception for learning meaningful food representations in an implicit manner. To this end, we present SCONE, a food-scooping robot learning framework that leverages representations gained from active perception to facilitate food scooping policy learning. SCONE comprises two crucial encoding components: the interactive encoder and the state retrieval module. Through the encoding process, SCONE is capable of capturing properties of food items and vital state characteristics. In our real-world scooping experiments, SCONE excels with a 71% success rate when tasked with 6 previously unseen food items across three different difficulty levels, surpassing state-of-the-art methods. This enhanced performance underscores SCONE's stability, as all food items consistently achieve task success rates exceeding 50%. Additionally, SCONE's impressive capacity to accommodate diverse initial states enables it to precisely evaluate the present condition of the food, resulting in a compelling scooping success rate. For further information, please visit our website.

**Keywords:** Food Manipulation, Robot Scooping, Active Perception

## 1 Introduction

Robot scooping skills have the potential to greatly improve human life, with applications in food preparation and assistive feeding [1, 2, 3, 4, 5, 6, 7] applicable in diverse environments, including homes, hospitals, and restaurants. Nonetheless, successfully scooping various food types presents a challenging task because of the wide range of appearances and physical properties exhibited by different foods. These attributes are not easily defined or fully predicted beforehand during preparation. Consequently, developing a framework that empowers robots to acquire the essential skills for handling unfamiliar food items remains a substantial challenge in real-world applications.

Present methodologies [1, 2, 3, 7, 8, 9] often rely on hard-coded primitive actions to address real-world food manipulation challenges. Nonetheless, these approaches frequently hinge on predetermined trajectories or specialize in particular food categories, constraining their adaptability to unfa-

---

[*]Corresponding Author

7th Conference on Robot Learning (CoRL 2023), Atlanta, USA.

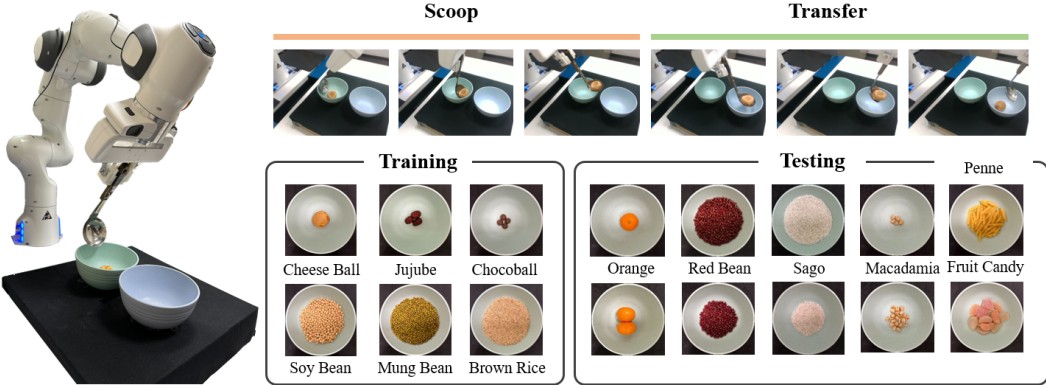

Figure 1: Real World Robot Food Scooping Task.

miliar food items. Current data-driven learning frameworks for assistive feeding [4, 10, 11, 12, 13] presume that food items with similar visual appearance will yield comparable success rates when adopting identical policies. However, due to the diverse properties of food encountered in scooping tasks, these frameworks still face limitations in generalizing their performance. Moreover, intricate characteristics of food states pose difficulties in precisely defining segmentation masks or bounding boxes as representations [4, 11, 12, 13].

Dynamic feedback from interactions could effectively capture essential food state features without explicit definitions. Hence, active perception [14, 15, 16] show great potential in tackling real-world food manipulation's generalization challenges by recognizing object characteristics through interaction. These frameworks consist of two stages. The first is the *Interacting* stage, where the robot interacts with target objects to acquire dynamic sensory feedback. Subsequently, in the *Manipulation* stage, a policy model benefits from the additional information gathered during the first stage, leading to improved efficacy. The pipeline has been widely used in food classification, policy selections [17, 18, 19], and dynamics inference [20, 21, 22]. Compared to static observations, interacting with food items yields higher confidence in assessing food properties.

To this end, we propose SCONE: a food SCOoping robot learNing framEwork with active perception. We employ active perception to enhance the manipulation of target objects in a coarse *plus* fine manner: it involves dynamics feedback for global food properties and local state information during each *Interacting* frame. Building upon this intuition, our method encodes active perception data with two distinct modules. The interactive encoder encodes overall interaction observations, while the state retrieval module captures crucial state details. Subsequently, a task-related embedding is obtained by fusing these two encoding modules. This embedding, along with current observations, serves as guidance for the closed-loop policy.

We evaluate our method on a real-world robotic food scooping platform, demonstrating its compelling performance compared to the benchmark methods. Overall, our method attains an impressive 71% task success rate across 6 previously unobserved food categories spanning three distinct levels of difficulty, thereby confirming its generalization capabilities. Furthermore, SCONE demonstrates its ability to accurately perceive current food states, ensuring continuous scooping success while minimizing spillage and damage to the food and surroundings. Additionally, we visualize the learned latent embeddings from the feature extraction modules, demonstrating that the properties of food can be extracted and the task-related information can be retrieved at each time step. Our main contributions are as follows:

- We propose SCONE: An active perception-based learning framework for food scooping robots, employing *Interaction* data to enhance generalization concerning food properties in real-world food scooping tasks.

- The visualization of latent embeddings highlights the extraction of valuable knowledge and clearly illustrates how active perception aids in policy learning.

- In real-world experiments, the framework effectively manipulates unseen food items across various settings, surpassing the performance of the baseline methods.

## 2 Related Work

**Robotic Food Scooping.** Recent studies have explored various robot scooping methods for handling food items [23, 24]. However, achieving effective food scooping requires careful consideration of the food's inherent characteristics, such as fragility, to enable a successful scooping policy. For assistive feeding and food preparation tasks, several commercial meal-assistance robots [8, 9] rely on pre-defined trajectories and need human-assigned controls, which are more stable but with limited autonomy. Ohshima et al. [1] introduce an autonomous robot system using Laser Range Finder for non-rigid solid foods by tackling the recognition problem of remnants of foods. However, it is still limited to the geometry and deformability of food items. Park et al. [2] presented an active feeding framework where users select tool-use actions. Without considering the properties of each food, this may encounter limitations in scalability. Grannen et al. [4] proposed a bimanual scooping framework with closed-loop visual feedback to prevent food breakage. However, the visual detection mechanism with masks in their framework still limits the range of food states. In this work, we explore the potential of active perception to represent foods in an implicit manner, offering an alternative approach for generalizing to diverse states of food.

**Active Perception in Closed-loop Policy Learning.** Prior works have utilized pre-interacting data for predicting object physical properties, reconstructing the object geometries, and executing open-loop control for manipulation tasks [25, 17, 26, 18, 27, 19]. However, it remains largely unexplored how the pre-programmed interaction can improve closed-loop policy learning in an implicit manner. Saito et al. [21] propose a sensorimotor dynamical system model using a recurrent neural network to learn the pouring task. In testing, robots observe liquid behavior with pre-programmed shaking actions and generate pouring motions. In their subsequent work, Saito et al. [22] employ a similar architecture for tool selection and tool-use tasks, enhancing the dynamical model with a multiple-timescale recurrent neural network. This network has updatable initial parameters, allowing exploration during interaction in testing. In contrast, we take a different approach to enhance the connection and use of interaction data in manipulation. We introduce the state retrieval module, which efficiently extracts crucial information from each step in the closed-loop process, allowing for more effective utilization of the interaction data.

## 3 Method

Our goal is to design a learning framework that enables real-world robot scooping. We develop our method based on active perception and focus on how to utilize the interaction data to improve task success under dynamic situations. Moreover, we adopt the learning from demonstration (LfD) paradigm to enable the robot to learn using a spoon in a faster and more reliable manner. We assume a demonstration dataset is available for model training. Figure 2 shows an overview of our proposed framework.

### 3.1 Problem Formulation

**Task Definition.** In our work, the food scooping task is divided into three distinct stages: the interacting stage, the scooping stage, and the transferring stage. During the interacting stage, the robot employs a spoon to stir the contents of the container while observing the dynamics of the food items. In the scooping stage, the robot's objective is to scoop up the food items successfully. Finally, in the transferring stage, the robot transfers the scooped food to another container. In this work, we focus on learning the challenging scooping process exclusively, while keeping the interacting and transferring stages fixed with pre-defined motions. Task success is determined by specific criteria:

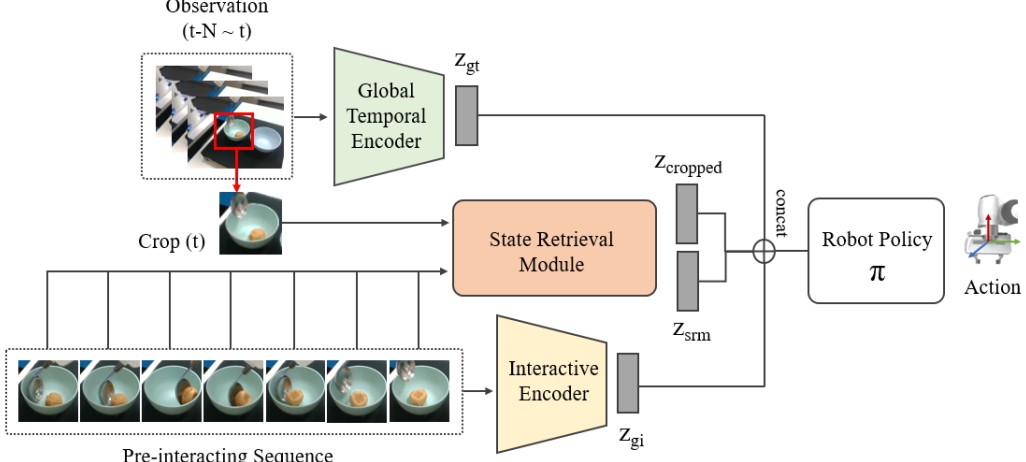

Figure 2: **SCONE** is a real-world robotic food scooping learning framework. With active perception, the generalization capability of the model can be enhanced. SCONE consists of three main feature extraction modules: (a) **Global temporal encoder**, which is able to capture the stage information from the current observations. (b) **Interactive encoder**, which effectively gathers information and integrates it to form a holistic understanding of the physical properties of the food items. (c) **State retrieval module**, which explores and extracts the task-related state information based on the current state to facilitate manipulation.

the food items on the spoon should cover over a one-third area or more than one entity without spillage and damage. The spoon must remain at a position higher than the container after scooping motions are performed.

**Pre-processing of Interaction Data.** To extract the information of food items with greater precision and efficiency, we sample K frames (K = 7 in our case) of observed data from sequences received in the interacting stage $D_{\texttt{interact}}^{\texttt{origin}} = \{(o_i, e_i)|i = 1, \ldots, N\}$. We also crop the viewing area around the container. The sampling starts at the time step when the spoon reaches the lowest position and acquires the data every 0.5 seconds. The pre-processed data denotes $D_{\texttt{interact}} = \{(o_i, e_i)|i = 1, \ldots, K\}$, which has sensor readings $o_i = (I_{\texttt{RGB}}, I_{\texttt{Depth}})$ and the robot end-effector poses $e_i$.

**SCONE Architecture (Fig. 2).** At each time step, we have the interaction data $D_{\texttt{interact}}$ and the N-length history of observation data $O_t = \{(o_i, e_i)|i = t - N, \ldots, t\}$. They are the inputs to the three encoding modules: global temporal encoder $f_{\texttt{gt}}$, interactive encoder $f_{\texttt{gi}}$, and the state retrieval module $f_{\texttt{srm}}$ to obtain latent vectors $z_{\texttt{gt}}, z_{\texttt{cropped}}, z_{\texttt{gi}}$ and $z_{\texttt{srm}}$. We aim to learn the policy $\pi(a_i|z_{\texttt{gt}}, z_{\texttt{cropped}}, z_{\texttt{gi}}, z_{\texttt{srm}})$ that minimizes the standard policy learning loss. In this work, we only control the 3D position and the y-axis pitch rotation of the end-effector, and the action $a_i \in R^4$ output from the policy at each time step is formulated as the delta between the current and next end-effector pose, i.e., $e_i$ and $e_{i+1}$.

## 3.2 Global Temporal Encoder

The aim of the global temporal encoder is to capture the progressive information from the observations, guiding the policy to be aware of the current phase in the task progress. We use two designed convolution neural networks to extract feature vectors of a certain dimension from the overall observations of $I_{\texttt{RGB}}$ and $I_{\texttt{Depth}}$. To obtain the temporal information, we encode the past $N$ frames of observations ($N = 10$ in our case), so at each time step, the inputs to the observation encoding module are $o_{\texttt{observe}} = (I_{\texttt{RGB}} \in R^{N \times W \times H \times 3}, I_{\texttt{Depth}} \in R^{N \times W \times H \times 1})$. These two sequences of different modalities pass through the corresponding encoders $f_{\texttt{RGB}}$ and $f_{\texttt{Depth}}$ to obtain two sets of feature embeddings. We concatenate them on the last dimension. Then, we flatten and downsample the feature vectors through one fully-connected layer, obtaining global temporal features of current observation $z_{\texttt{gt}}$. This would be the overall guidance for the policy model to output actions.

### 3.3 Interactive Encoder

To extract a representation of food items through pre-programmed interaction, we introduce the interactive encoder $f_{gi}$, which consists of an RGB encoder, a depth encoder, and a self-attention module. The sequential data of two modalities $o_{\mathtt{interact}} = (I_{\mathtt{RGB}} \in R^{K \times W \times H \times 3}, I_{\mathtt{Depth}} \in R^{K \times W \times H \times 1})$ are first passed over the convolution encoder separately, and then concatenated into latent vectors $z_{\mathtt{interact}} \in R^{K \times D}$. Afterward, the concatenated embeddings are fed into the self-attention module, implementing cross-modality and cross-time attention to catch the critical information, outputting final flattened latent vectors $z_{gi}$. The $z_{gi}$ contains the characteristics of food items in the latent space that can be treated as guidance for the policy model to recognize the similarity between seen and unseen objects.

### 3.4 State Retrieval Module

We assume that the state of each frame in the interacting stage has a different importance to the state of each time step in the manipulating stage, so the objective of the state retrieval module (SRM) is to retrieve meaningful embeddings from the sequences of interaction data queried on current observations. To obtain meaningful features for guidance, the state feature extraction layers and the cross-attention mechanism are applied.

To extract the features more precisely, we crop the current observation around the container to obtain $o_t^{\mathtt{crop}}$, which is the same as the pre-processing of the interacting data. First, the RGB-D images in $o_t^{\mathtt{crop}}$ are fed into designed convolution layers to output the feature embeddings separately, and then concatenated with the corresponding end-effector poses, acquiring the state pairs $z_{\mathtt{cropped}} \in R^{1 \times D}$ and $z_{\mathtt{interact}} \in R^{K \times D}$. Next, the attention mechanism is applied to generate an attention map between $z_{\mathtt{cropped}}$ and $z_{\mathtt{interact}}$. The final feature maps are the result of matrix multiplication of the attention map and each frame features in $z_{\mathtt{interact}}$. Furthermore, a fully connected layer is used to downsample the output latent vectors into lower dimensions. The output of the state retrieval module $z_{\mathtt{srm}}$ extracts task-related state information from the pre-interacting data, and implicitly guides the policy model to adapt the policy considering the target item and the interaction experience. Concurrently, the $z_{\mathtt{cropped}}$ are treated as another input to the policy model.

### 3.5 Policy Learning

To learn the policy $\pi(a_i | z_{\mathtt{gt}}, z_{\mathtt{cropped}}, z_{gi}, z_{\mathtt{srm}})$, we adopt Behavior Cloning (BC) to train action prediction from the observation in a supervised manner. The model takes the latent vectors of the global observation, the local observation, the overall food representation, and the retrieved cue under the current state as inputs pass over a three-layer Multi-Layer Perceptron, and then outputs the corresponding actions. We use the mean square loss between the ground truth action $\hat{a}$ and predicted action $a$ for training the network.

## 4 Evaluation

### 4.1 Food Scooping Dataset

**Hardware setting.** The data collection platform consists of a 7DoF Franka Emika Panda robot with the default gripper, a D435 RealSense camera fixed around the platform, and two containers placed in the center of the table. During the data collection, we receive sensory data of RGB-D images and the robot proprioception simultaneously. The robot controller runs at a 10 Hz frequency.

**Food Categories.** In the training stage, we select 6 food items, i.e., brown rice, mung bean, soybean, chocolate ball, dried jujube, and cheese ball) that encompass different particle sizes, shapes, and weights. Additionally, we evaluate 6 unseen food items, i.e., orange, red bean, sago, macadamia, penne, and fruit candy. To assess the generalization ability of our proposed method, we design three difficulty levels. The *basic* configuration closely mimics the training data, with the container being filled with two-thirds full of food items. These primarily comprise small particles, accompanied by a

smaller quantity of larger particle foods. The *extended* setting reduces the number of small particles and increases the number of large particles. The *peculiar* setting includes foods with significantly different appearances and weights compared to those in the training set. These difficulty levels are designed to have a comprehensive assessment of the model's capability to handle a wide range of scenarios. Additionally, we have introduced testing scenarios that involve foods with notably different properties, such as semi-solid food and liquid. See Supplementary Material for the details.

**Data Collection.** We employ the hand-guiding mode for intricate scooping operations. This involves a human kinesthetically guiding the robot while recording the end-effector's trajectory. We gather 10 demonstrations for each food category, each encompassing 6 seconds of interaction, 15 seconds of scooping, and 10 seconds of transferring. The recorded interacting motions are subsequently replayed to collect demonstrations without humans in the scene. Every trial is thoroughly reviewed to ensure trajectory viability. A trial is considered a qualified demonstration if the robot successfully scoops up the food items. In total, we collect 60 real-world demonstrations.

## 4.2 Baselines

We conduct a comparison with seven different approaches, which encompass various BC-based methods and template-based methods. Please see Supplementary Material for the details of each baseline.

**BC-based Methods:** The methods include the standard BC trained on both scooping and transferring data (**one stage**), standard BC trained w/o interaction data, BC with interaction, and the BC conditioned on food categories represented by a one-hot vector.

**Template-based Methods:** The method treats the learning process as a template selection problem. The category-level templates are obtained through the mean of all trajectories that belong to the same category. The experiments proceed with selecting templates **randomly** or **pre-defined** with the trained classifier.

**MTRNN [22]:** The baseline strings the interacting and the scooping stage as one action sequence. The model trains a multi-timescale recurrent neural network to learn the corresponding system.

## 4.3 Single Scooping Experiment

In the single scooping task, the robot is required to scoop up food items and transfer them to the target container without causing spillage or damaging the food. The food items on the spoon should occupy at least one-third of its area. All tested foods are uniformly spread in the bowl, ensuring similar initial conditions for all testing trials. The experimental results are presented in Table 1. Overall, our proposed method achieves a task success rate of 71% across 3 difficulty levels.

**Basic Setting.** BC baselines (**one stage** and **w/o interact**) perform unsatisfactory in this setting. The incorporation of additional information from interacting data leads to improved performance, with a success rate increase of over 10% compared to the basic BC baselines. Template-based methods offer improved stability over BC-based methods, yet there are cases where classifier predictions may lead to the selection of an unsuitable template, resulting in failure instances. For instance, when encountering the food item **Sago**, the model predicts incorrectly, leading to food spillage. The MTRNN and the SCONE achieve over 80% success rate in each category, while there are failure cases caused by unstable robot arm movements or other sources of uncertainties.

**Extended Setting.** The overall task success rate declines as a result of disparities between the testing and training environments, posing challenges in accurately assessing the states of the target items and leading to failed attempts at food scooping. Conversely, our proposed framework excels at leveraging the precise state information obtained during the interaction stage, resulting in improved performance. The template-based method (Classified Template) maintains stability when the appropriate template is selected.

| | Basic | | | | Extended | | | | Peculiar | | |
|---|---|---|---|---|---|---|---|---|---|---|---|
| | Orange | Red Bean | Sago | Macadamia | Orange | Red Bean | Sago | Macadamia | Penne | Candy | **Success** |
| BC (one stage) | 1/10 | 0/10 | 0/10 | 1/10 | 0/10 | 0/10 | 0/10 | 0/10 | 1/10 | 0/10 | 3% |
| BC (w/o interact) | 3/10 | 0/10 | 2/10 | 0/10 | 5/10 | 0/10 | 3/10 | 2/10 | 2/10 | 4/10 | 21% |
| BC (w/ food id) | 7/10 | 2/10 | 0/10 | 2/10 | 0/10 | 0/10 | 3/10 | 0/10 | 4/10 | 3/10 | 21% |
| BC (w/ interact) | 7/10 | 5/10 | 0/10 | 3/10 | 3/10 | 0/10 | 3/10 | 1/10 | 5/10 | 1/10 | 28% |
| Rand. Template | 5/10 | 4/10 | 6/10 | 8/10 | 0/10 | 0/10 | 3/10 | 0/10 | 3/10 | 0/10 | 29% |
| Classified Template | **10/10** | **10/10** | 0/10 | **10/10** | 3/10 | 0/10 | **10/10** | 3/10 | 1/10 | 0/10 | 47% |
| MTRNN [22] | **10/10** | **10/10** | **10/10** | 8/10 | 4/10 | 2/10 | 0/10 | 3/10 | **8/10** | 3/10 | 58% |
| SCONE (Ours) | **10/10** | 8/10 | 9/10 | 9/10 | **7/10** | **5/10** | 6/10 | **5/10** | 7/10 | **5/10** | **71%** |

Table 1: Single Scooping Experiment Results.

| | Penne | | | Red Bean | | |
|---|---|---|---|---|---|---|
| | AS↑ | Max/Min | Penalty↓ | AS↑ | Max/Min | Penalty↓ |
| Classified Template | 0 | 0/0 | 0% | 5.4 | 8/2 | 29% |
| MTRNN [22] | 2.8 | 4/2 | 42% | 2.8 | 4/2 | 21% |
| SCONE (Ours) | 3.2 | 4/2 | 25% | 13.4 | 14/5 | 22% |

Table 2: Continuous Scooping Experiment Results.

| Global Temporal Encoder | Interactive Encoder | Local Latent | SRM | Success Rate |
|---|---|---|---|---|
| ✓ | | | | 21% |
| ✓ | ✓ | | | 28% |
| ✓ | ✓ | ✓ | | 48% |
| ✓ | ✓ | ✓ | ✓ | 71% |

Table 3: Ablation Study.

**Peculiar Setting.** The task success rate of template-based methods significantly decreases in comparison to previous settings, highlighting the effectiveness and robustness of the close-loop learning-based approach. The methods with interaction outperform other baselines because of their enhanced understanding of food items. SCONE achieves the highest total success rate, which can be attributed to its ability to accurately capture task-related information and establish an effective connection between the pre-interaction state and the subgoal state during the manipulation stage.

## 4.4 Continuous Scooping Experiment

In the continuous scooping task, the robot needs to repetitively transfer food from one bowl to another until it scoops up nothing. Unlike the single scooping task, a model should precisely perceive food properties and states to achieve continuous scooping with changing initial conditions. A notable change from the single scooping experiment is the removal of the criterion involving scooping less than one-third of a spoon as a failure. We conduct a comparison between the proposed method and other representative baselines in scooping tasks involving red beans and penne pasta, with five trials for each type of food. SCONE demonstrates the ability to successfully scoop food across intricate variations in food states, with a lower proportion of spillage and damage to the food and surroundings. The experimental results are presented in Table 2. **AS** means the average number of successful scoops per trial, **Max/Man** represents the maximum and minimum number of scoops in a single trial, and the **Penalty** shows the total instances of food spillage and food/setting damage, divided by the total number of scoops.

## 4.5 Ablation Study

We compared four model variants in the single scooping experiment, as shown in Table 3. The base model achieves a success rate of 21%. Incorporating interactive data improved performance to 28%, indicating a better understanding of the target food items. Moreover, adding the current local observation, which emphasizes the model's attention on the crucial stage, results in a substantial

improvement of approximately 20% in the task success rate. Finally, including the state retrieval module significantly enhanced the success rate to an impressive 71%, showcasing its ability to accurately extract and utilize critical information from the interacting data.

## 4.6 Visualization

**Interactive Embeddings.** To interpret the behavior of the interactive encoder, we utilize t-SNE projections to visualize the embeddings from the interactive encoder. In Figure 3, we show that food items belonging to the same category or possessing similar properties tend to cluster together. This result suggests that the interactive encoder effectively captures the inherent similarities among the food items.

**Attention Scores from State-retrieval Module.** Figure 4 depicts attention scores during manipulation. The x-axis represents the time index in the manipulation stage, and the y-axis shows attention scores for state pairs. The cross-attention mechanism effectively identifies and captures state importance within the interaction data, enhancing manipulation. Attention scores

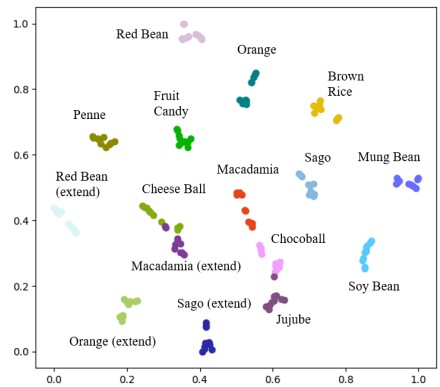

Figure 3: 2D t-SNE of Interactive Embeddings.

follow a consistent pattern across trials, showcasing the model's ability to understand task phases. Notably, even without explicit labeling, the model demonstrated an inherent ability to capture and recognize important information related to different phases of the task. Further analysis can be found in the Supplementary Material.

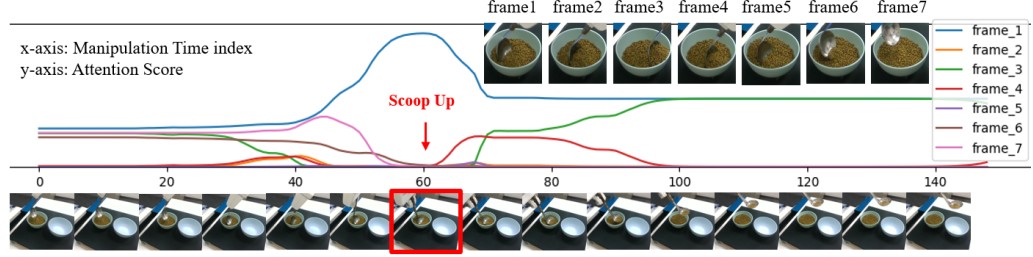

Figure 4: Visualization of Attention Scores.

## 5 Conclusion

We introduced SCONE, an active perception-based robot food scooping framework, utilizing interacting data in a coarse plus fine approach: global encoding for overall physical properties, and state retrieval for critical state information. Our framework dynamically tailors the scooping policy according to real-time observations, enabling the model to excel with previously unobserved food items, thereby elevating both performance and generalization capabilities.

**Limitations and Future Work.** Our proposed approach places significant emphasis on extracting vital information from active perception, making the design of pre-programmed interaction motions an essential component for attaining successful outcomes. However, the current approach of human-hard-coded-defined actions may limit the exploration of other possibilities. In our future endeavors, we envision ample room for further investigation into the design of interaction motions, aiming to enhance their exploratory nature, thereby augmenting the system's overall performance. In addition, our emphasis on food property-driven generalization imposes constraints on environmental variables such as container positions and tool types. Thus, our goal is to explore the integration of perception components, such as object detection and pose estimation, into the proposed framework, allowing it to adapt to a broader range of diverse setups.

**Acknowledgments**

The work is sponsored in part by the Higher Education Sprout Project of the National Yang Ming Chiao Tung University and Ministry of Education (MOE), the Yushan Fellow Program Administrative Support Grant, and the National Science and Technology Council (NSTC) under grants 110-2222-E-A49-001-MY3, 110-2634-F-002-051, 111-2634-F-002-022-, and Mobile Drive Technology Co., Ltd (MobileDrive).

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

# SCONE: A Food Scooping Robot Learning Framework with Active Perception Supplementary Material

The supplementary materials consist of:

1. Demonstration video.
2. Details of dataset.
3. Baseline approach.
4. Analysis of experiment results.
5. Details of our proposed method.
6. Visualization of attention scores.
7. Additional experiment on out-of-distribution food categories.

## 1   Demonstration video

In the supplementary video, we show 1) a brief introduction of our food scooping robot learning framework and 2) illustrations of scooping tasks and qualitative results.

## 2   Details of Dataset

In this section, we provide more details about the data collection and preparation process for training.

### 2.1   Food Preparation

We select a total of 12 categories of food items for our real-world experiment, with 6 categories used for training and the remaining 6 categories used for testing. To simplify the complexity, we have limited the differences between categories primarily to **particle size** and the **amount** of food items.

**Food in the Training Set.** The food categories for training contain brown rice, mung bean, soybean, chocolate call, dried jujube, and cheese ball. For food items with small particles such as brown rice, mung bean, and soybean, we fill the bowl to approximately 2/3 of its capacity. This threshold ensures that the interaction can be carried out without the risk of spilling the food items. For food items with large particles such as chocolate balls, dried jujube, and cheese balls, we select more than one piece but fewer than a certain number based on their size. This is done to prevent the spoon from getting stuck or breaking the food items when there is insufficient space in the bowl for the spoon to reach them.

**Food in the Testing Set.** The food categories for testing contain sago, red bean, orange, macadamia, penne, and fruit candy. In the testing set, we have designed three levels of difficulty to evaluate the performance of the models. Both the **Basic** and **Extended** settings in the evaluation include sago, red bean, orange, and macadamia, and the peculiar setting includes penne and fruit candy.

• **Basic Setting:** In this setting, the conditions are kept identical to the training set. For food items with small particles, the bowl is filled to approximately 2/3 of its capacity. For food items with large particle sizes, more than one entity is included in the bowl.

• **Extended Setting:** In this setting, we change the combination of properties related to particle size and amount of food items. We want to explore different scenarios to evaluate the performance of our model under varying conditions. For food items with small particles, we fill the bowl with a smaller

quantity of these items. The intention is to keep the height of the food in the bowl similar to that of the food items with large particles in the **Basic** setting. Conversely, we increase the amount of food items with large particles.

• **Peculiar Setting:** These food items in the peculiar setting had unique features, such as different shapes, colors, or textures, that are not present in the training set. By introducing these visually distinct food items, we aim to challenge the model's capacity to recognize and handle novel objects effectively.

## 2.2  Manipulation Policy

During the data collection phase, we employ two distinct manipulation policies to ensure the successful scooping of the food items. To prevent spilling, we adopt a specific policy for scooping up food items with small particles. The strategy involves positioning the spoon at a shallow depth under the height of the food items. By adopting this approach, the risk of spillage can be minimized and the food items were securely contained within the spoon during the scooping process. When dealing with food items that have large particles, we position the spoon at the lowest point within the bowl and gently push the items toward the edge. This allows the food to roll into the spoon and is able to successfully scoop up without any spillage. Figure 1 shows the end-effector positions during manipulation in the training dataset.

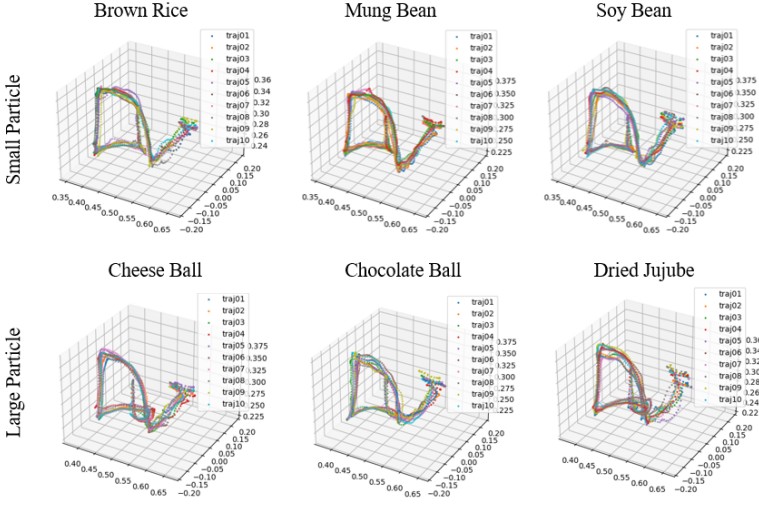

Figure 1: Visualization of End-effector Position During Manipulation.

## 2.3  Implementing Detail of Overall Scooping Task

The task is divided into three stages: interacting, scooping, and transferring. Among these stages, only the scooping stage requires learning. The overall task follows a predefined procedure. The interaction data is collected by replaying the recorded end-effector trajectory. The model then begins predicting the scooping stage. Once the scooping stage is completed, the transferring stage is initiated. The trajectory for the transferring stage is obtained by averaging all training data. Considering that the ending position of the scooping stage for every testing food tends to be relatively close to initial position of the transferring stage, the Panda arm then proceeds to move to the first point using simple motion planning. Once the transition is complete, the rest of the trajectory is replayed accordingly.

## 3  Baseline Approach

This section provides detailed explanations of the baselines in comparison with SCONE.

## 3.1 BC-based Method

When adopting Learning from Demonstration (LfD), behavior cloning (BC) is a straightforward approach for direct learning from both observation and action. We conducted the task using 4 different BC models, both with and without the inclusion of interacting data.

**BC one stage.** In BC (one stage), we consider scooping and transferring as a single stage. This implies that the BC (**one stage**) model is required to learn the long-horizon task only through observation, without any additional input.

**BC without interaction.** The most basic method that learns the scooping task through observations.

**BC with food ID.** To implement BC with food ID, we applied the trained classification model to the interaction data. The predicted food ID was then conditioned as a one-hot vector and concatenated with the observation $z_{\tt gi}$, serving as the input for the BC model.

**BC with interaction.** In the case of BC with interaction, the input for the model is the concatenation of latent of observation and interaction data.

## 3.2 Template Policy

One of the approaches we explore in utilizing interaction data is the selection of trajectories from the pre-interacting data based on their similarity to the food categories in the training dataset. The template trajectory is obtained by averaging the sequences of end-effector pose of 6 different food categories respectively.

**Rand. Template.** Random policy is selected arbitrarily.

**Classified Template.** The template policy is selected accordingly based on the predicted category by the classifier.

## 3.3 Dynamical System Model

**MTRNN.** We utilized the multiple time scale recurrent neural network by [1] to update the initial parameter $Cs_0$ using the interaction data in the testing stage.

# 4 Analysis of Experiment Results

## 4.1 Failure Cases

During our real-world evaluation, the failure cases observed included instances of spilling (SP), insufficient food on the spoon (IF), failed attempts to scoop (FA), collisions (CO), and others (OT). We select the orange, sago, and penne as examples, and their results can be found in Table 1 and Table 2.

**Spilling (SP).** Spilling happens at the end of the scooping process, especially when there is an excessive amount of food on the spoon. Consequently, during the stages of transferring or when withdrawing the spoon from the bowl, the excess food spills out.

**Insufficient food (IF).** The failure case of insufficient food occurs when the amount of food on the spoon is unable to cover at least one-third of its surface area. This is frequently attributed to the spoon not being inserted deep enough into the bowl to effectively reach the desired food portion.

**Failed Attempts (FA).** Failed attempts to scoop are attributed to the same underlying reason as insufficient food. In these cases, the spoon fails to acquire any amount of food, rather than scooping up an insufficient quantity.

**Collisions (CO).** Relying on vision-based information can lead to incorrect decisions when facing out-of-distribution situations. In such scenarios, collisions between the spoon and the bowl may occur.

| | SP | IF | FA | CO | OT | Failed |
|---|---|---|---|---|---|---|
| | **Basic - Orange** | | | | | |
| BC (one stage) | 0 | 0 | 5 | 3 | 1 | 9 |
| BC (w/o interact) | 2 | 0 | 3 | 3 | 0 | 8 |
| BC (w/ food id) | 1 | 0 | 2 | 1 | 1 | 5 |
| BC (w/ interact) | 1 | 0 | 2 | 0 | 0 | 0 |
| Rand. Template | 0 | 0 | 5 | 0 | 0 | 5 |
| Classified Template | 0 | 0 | 0 | 0 | 0 | 0 |
| MTRNN [1] | 0 | 0 | 0 | 0 | 0 | **0** |
| SCONE (Ours) | 0 | 0 | 0 | 0 | 0 | **0** |

(a) Basic - Orange

| | SP | IF | FA | CO | OT | Failed |
|---|---|---|---|---|---|---|
| | **Basic - Sago** | | | | | |
| BC (one stage) | 0 | 0 | 0 | 0 | 10 | 10 |
| BC (w/o interact) | 8 | 0 | 0 | 0 | 0 | 8 |
| BC (w/ food id) | 1 | 0 | 2 | 1 | 1 | 5 |
| BC (w/ interact) | 2 | 0 | 2 | 8 | 0 | 10 |
| Rand. Template | 2 | 0 | 2 | 0 | 0 | 4 |
| Classified Template | 10 | 0 | 0 | 0 | 0 | 10 |
| MTRNN [1] | 0 | 0 | 0 | 0 | 0 | **0** |
| SCONE (Ours) | 1 | 0 | 0 | 0 | 0 | 1 |

(b) Basic - Sago

Table 1: Basic - Failure Cases

| | SP | IF | FA | CO | OT | Failed |
|---|---|---|---|---|---|---|
| | **Extended - Sago** | | | | | |
| BC (one stage) | 0 | 0 | 0 | 0 | 10 | 10 |
| BC (w/o interact) | 5 | 1 | 0 | 1 | 0 | 7 |
| BC (w/ food id) | 1 | 0 | 2 | 1 | 1 | 5 |
| BC (w/ interact) | 0 | 4 | 0 | 3 | 0 | 7 |
| Rand. Template | 3 | 4 | 0 | 0 | 0 | 7 |
| Classified Template | 0 | 0 | 0 | 0 | 0 | **0** |
| MTRNN [1] | 0 | 5 | 0 | 5 | 0 | 10 |
| SCONE (Ours) | 3 | 1 | 0 | 0 | 0 | 4 |

(a) Extended - Sago

| | SP | IF | FA | CO | OT | Failed |
|---|---|---|---|---|---|---|
| | **Peculiar - Penne** | | | | | |
| BC (one stage) | 1 | 0 | 8 | 3 | 0 | 9 |
| BC (w/o interact) | 1 | 0 | 6 | 1 | 0 | 8 |
| BC (w/ food id) | 0 | 0 | 5 | 1 | 0 | 6 |
| BC (w/ interact) | 1 | 0 | 0 | 4 | 0 | 5 |
| Rand. Template | 4 | 0 | 3 | 0 | 0 | 7 |
| Classified Template | 2 | 0 | 7 | 0 | 0 | 9 |
| MTRNN [1] | 2 | 0 | 0 | 0 | 0 | **2** |
| SCONE (Ours) | 2 | 0 | 1 | 0 | 0 | 3 |

(b) Peculiar - Penne

Table 2: Extended and Peculiar - Failure Cases

**Others (OT).** In addition to the failure cases mentioned earlier, there are instances where the task cannot be successfully completed or the food ends up being damaged.

## 4.2 Result Analysis

**Table 1: Orange.** The BC-based baselines achieved low task success rates due to the high incidence of failed attempts because the weight of the orange used in the testing set is heavier than the foods in the training set. By contrast, the MTRNN and SCONE models exhibited stable performance due to their ability to learn and understand the conditions for successful scooping from the provided demonstrations. The classified template method also achieved higher performance due to its ability to select suitable templates for food items.

**Table 1: Sago.** The BC-based baselines encountered challenges in providing correct predictions, causing several failure cases during the manipulation. Moreover, the template-based method has been limited by its reliance on the predictions from the classifier. In cases where the classifier selected a template intended for food items with large particles, it often resulted in spillage. The MTRNN and the proposed SCONE method demonstrated their ability to overcome this challenge, achieving a higher success rate.

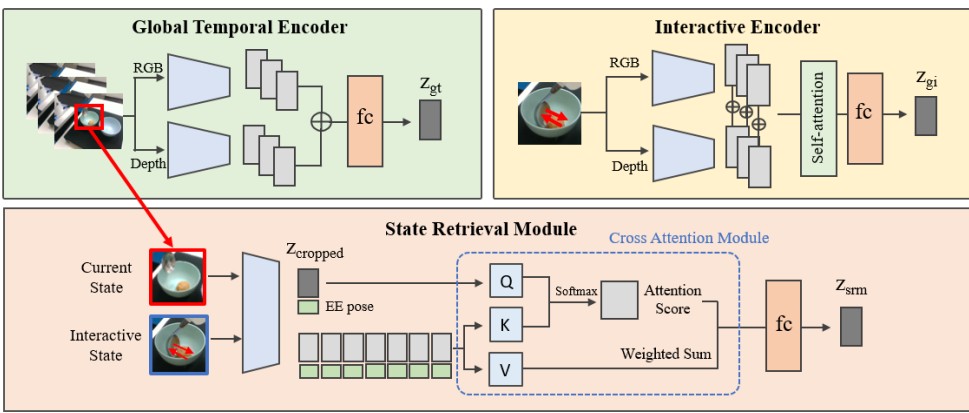

Figure 2: Details of Each Module in SCONE

**Table 2: Extended Sago.** The number of spillages (SP) decreases significantly compared to the basic settings because the amount of sago been reduced, but there was an increase in the occurrence of insufficient food (IF) cases, due to the improper depth insertion of the spoon into the bowl in most of the methods. The Select Template model classified sago as chocolate balls, resulting in 10 successful trials when following the corresponding template. MTRNN demonstrated poor performance under this particular setting. During the scooping stage, it showed a hovering behavior within the bowl and occasionally collided with it. However, SCONE is capable of handling the challenging setting, achieving a success rate of 6 out of 10.

**Table 2: Penne.** To test the models' generalization abilities, we conducted experiments using peculiar-shaped foods such as penne and fruit candy. Though the color of penne looks familiar to soy beans, the shapes of them are totally different, which led to failed attempts (FA) when employing the soy bean template. While MTRNN was able to handle penne, the jittering trajectory could lead to instability and spillage (SP), which we would like to avoid in real-world evaluation. Our SCONE behaved more stable and maintained a smooth trajectory.

# 5   Details of Proposed Method

See Figure 2 for detailed information on each encoding module. All the observation inputs are RGB-D images, which are processed through corresponding convolutional layers to extract features.

**Global temporal encoder.** The global temporal encoder takes as input a sequence of current observations of length N. In our implementation, we set N to 10, which means that the model can access the previous 10 observations within a time window of 1 second. This allows the model to capture the temporal dynamics and dependencies in the input data. Then, the sequences of RGB and depth images are processed separately by their respective encoders, and the output features are concatenated and flattened into a one-dimensional embedding. To further reduce the dimensionality of the features, a fully-connected layer is applied to downsample the features to the dimension of 128.

**Interactive encoder.** We utilize an interactive encoder to process the sequence of observations captured during the interaction stage. The number of frames K is set to 7. Similar to the encoding process in the global temporal encoder, both RGB and depth images within the sequence are processed independently through their respective convolutional layers. The output features from these layers are then concatenated to form a sequence of feature maps, capturing the visual information from both modalities. Then, we introduce the multi-head self-attention mechanism, allowing the model to focus on relevant parts of the input sequence. The output features passed through a fully-connected layer, which reduces their dimension to 128.

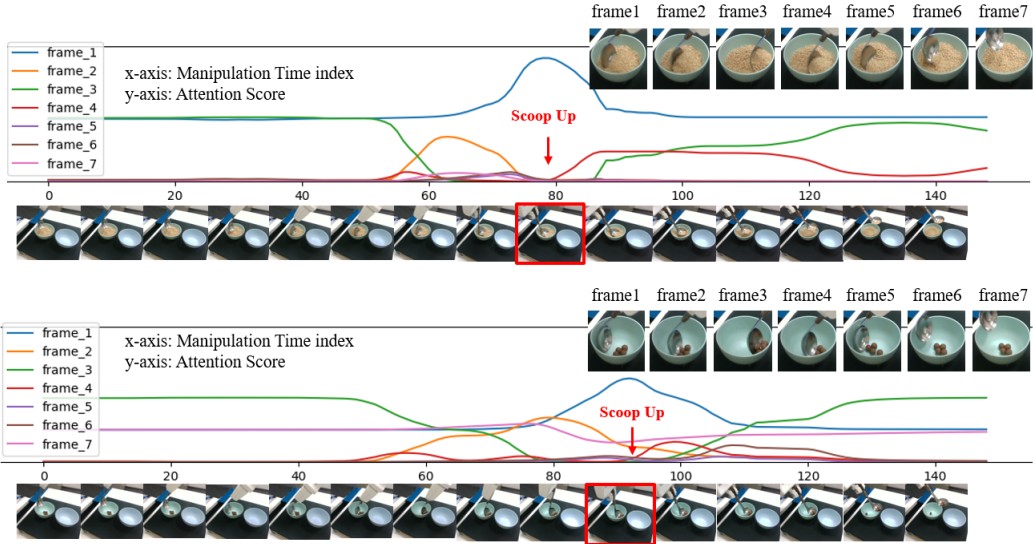

Figure 3: Visualization of Attention Scores. The upper is the manipulation process of brown rice, and the lower is the manipulation process of chocolate ball.

**State retrieval module.** The state retrieval module tasks the current local observation and the sequence of interaction as inputs. For the current local observation, which consists of cropped RGB-D images captured at the current time step, we pass them through an encoding module to obtain latent embeddings. These embeddings are then concatenated with the current end-effector pose, resulting in the latent state representation denoted as $z_{\texttt{cropped}}$. Regarding the sequence of interaction, we apply the same encoding layers to process the observations and obtain corresponding feature embeddings. Similar to the current local observation, we also concatenate the end-effector poses with the encoded features, creating a sequence of states in the interaction. To retrieve the critical state information, we use $z_{\texttt{cropped}}$ as a query and compute its relationship with the sequence of states in the interaction. This is achieved by employing cross-attention mechanisms that output weighted feature embeddings. These embeddings are then flattened and downsampled to a dimension of 32, obtaining the $z_{\texttt{srm}}$. Furthermore, we also downsample the latent state representation $z_{\texttt{cropped}}$ to a dimension of 32.

# 6  Visualization of Attention Scores

Figure 3 shows more examples of visualization for attention scores. Based on the results, it is evident that the model can accurately capture the state information without human labeling; this is demonstrated by the consistent patterns observed in the changing attention scores over time at each trial. Overall, our analysis reveals that **frame 1** and **frame 3** exhibit higher attention scores compared to other frames. We attribute this to the presence of important state information related to the spoon's contact with the food items under specific end-effector poses. These frames can be seen as providing subgoal-like cues, guiding the model in the scooping task. Additionally, we observed that in food items with large particles, **frame 3** tends to have higher attention scores than **frame 1** initially. This is because the state captured in **frame 3** is more similar to the state prior to the scooping action in scenarios involving large particles. On the other hand, we noticed that the highest attention score for **frame 1** typically occurs close to the timing of the actual scooping action. This suggests that **frame 1** serves as a general guide for determining "how" and "where" to scoop up the food items. These findings highlight the model's ability to effectively identify and utilize critical state information from the interacting data to inform its scooping strategy.

## 7 Additional Experiment on Out-of-distribution Food Categories

We conducted additional tests to assess the model's adaptability to food categories with markedly different physical properties compared to the training data. Specifically, we prepared semi-solid pudding and liquid black tea. The proposed method demonstrated a success rate of over 50% for both foods, indicating a notable level of generalization in our approach.

|  | Black Tea | Pudding |
| --- | --- | --- |
| SCONE (Ours) | 8/10 | 5/10 |

Table 3: Out-of-Distribution Food Scooping Experiment

## References

[1] N. Saito, T. Ogata, S. Funabashi, H. Mori, and S. Sugano. How to select and use tools?: Active perception of target objects using multimodal deep learning. *IEEE Robotics and Automation Letters*, 6(2):2517–2524, 2021.

