# OpenReview forum: "SCONE: A Food Scooping Robot Learning Framework with Active Perception"
_robot-learning.org/CoRL/2023/Conference — CoRL 2023 Poster_

### Official Review · Reviewer_VGVQ · 2023-06-23

**Confidence:** 4
**Originality:** Very Good
**Technical Quality:** Fair
**Clarity Of Presentation:** Very Good
**Impact:** 3

**Recommendation:**

Strong Accept: I recommend accepting the paper and will argue for my recommendation even if other reviewers hold a different opinion.

**Review:**

This work seems relatively original, where in the food manipulation space, it seems to be one of the few to introduce closed-loop control during food manipulation.

The paper overall is technically sound. The only concern is there seems to be little explanation for failures and the authors seem to imply the failures in the basic setting are mainly due to unstable robot arm movements or other sources. If this is the case, it should be written more clearly. An explanation for the extended and peculiar setting failures is also needed.

The paper is clearly written and easy to understand. The results show SCONE outperforms most of the baselines and the idea of closed-loop food manipulation with an implicitly learned food representation is a unique contribution.

It may be interesting to see if similar foods take similar trajectories. One limitation I see is that the test food seemed similar to the train food. Chocoballs appear to be similar to macadamia nuts and red bean is similar to soy and mung bean. It would be nice to see how this would function for more varied food, such as lychee (spikey), cut banana (cylinder shape), boba (or something of a more spongy texture).

Another limitation in my view is all experiments were done with the same ladle. In the video at least, the “spoon” also appears large, not a normal bite size spoon. It would be good to show results on a few differently shaped spoons as the way the paper presents it, the utensil used for food manipulation should not affect the performance of the method.


**Quality Of The Limitations Section:**

Additional details required

**Questions For Rebuttal:**

1. During data collection, do you collect 1 human demo per food and then replay that same demo 10 times per food for data? Is there no variation, such as injected noise, in the trajectory when replayed?

2. In the ablation studies, how many trials were conducted per ablation?

3. In the basic setting, why does SCONE do worse than the Classified Template and MTRNN baselines?

4. The generalization claim seems unstable because as mentioned above, the train food seems very similar to the test food and it would be nice to have tested food with slightly different shapes and textures.

5. It would be good to have experiments with multiple different spoon shapes.


**Robotics Focus:**

Sufficient demonstration on hardware

**Summary Of Paper:**

This paper contributes SCONE, which is a robotic food scooping learned method that uses active perception to implicitly learn a latent space representation for food items. This is later accessed with the state retrieval module during manipulation. This paper claims that prior work is limited in that they use hard-coded primitives for food manipulation, but by implicitly learning a representation of food from human trajectories with the human manipulating the food, the model is able to learn food dynamics and alter the scooping manipulation as needed to successfully scoop.

**Summary Of Recommendation:**

The work is original and clearly presented. The main concerns for a weak accept over a strong accept is for the following two reasons:

1) Technical quality. The train and test food were very similar so I question the generalizability of the method to unseen food types. They all embodies either a large, spherical food or grains/beans. There was also no report on the number of trials conducted in the ablation study.

2) Limitations. The biggest limitation I saw is all experiments were done on the same "spoon," but it appears to be a very large ladle which makes the problem easier. To address this limitation, the authors should either add this in the paper or test or train&test on various spoons as the method seems to have no reliance on the spoon shape.

---

> ### Author Response · Authors · 2023-08-13
> **Response to Reviewer VGVQ**
>
> We thank the reviewer's valuable and helpful feedback. To further showcase the potential of our framework, we conducted new experiments involving continuous scooping and scooping of liquid/semi-solid foods. The experiment results and demo videos demonstrate the model's capabilities in scooping under complex food conditions, even with unique items like black tea and pudding. For additional information, please refer to the general response section.
>
> ---
>
> **Q1: Little explanation for failures.**
>
> **A1:** Thank you for bringing this to our attention. Due to limitations in the length of the paper, we faced challenges in including analyses of all failure cases in the main text. We have included failure analysis in the **supplementary material** and relevant issues discussions. For more details, please refer to Section 4 of the supplementary material.
>
> ---
>
> **Q2: Test food seemed similar to the train food. It would be nice to see how this would function for more varied food, such as lychee (spikey), cut banana (cylinder shape), boba (or something of a more spongy texture).**
>
> **A2:** This is indeed an exciting test. We investigate the proposed method's ability for more diverse material-wise generalization. In our experiments, we tested with penne(cylinder shape), fruit candy (oval), black tea(liquids), and pudding (semi-solid), which are pretty different from the training data, and all achieved a success rate of over 50%. The **demo video can be found [here](https://sites.google.com/view/corlscone/%E9%A6%96%E9%A0%81)**. We could also consider testing spikey or other uniquely shaped food items to evaluate the model's generalization capabilities further. Thank you for the suggestion.
>
> ---
>
> **Q3: During data collection, do you collect 1 human demo per food and then replay that same demo 10 times per food for data?**
>
> **A3:** We do not collect one trajectory and replay it 10 times. Instead, we gather 10 human demonstrations and perform 10 robot demos for each food item. Therefore, while the trajectories for the same type of food may be pretty similar each time, variations are present.
>
>
> ---
>
> **Q4: In the ablation studies, how many trials were conducted per ablation?**
>
> **A4:** It includes 100 trials for each ablation.
>
> ---
>
> **Q5: In the basic setting, why does SCONE do worse than the Classified Template and MTRNN baselines?**
>
> **A5:** Thank you for the question. In the basic setting, the states of all foods are closer to those in the training set. In such cases, the template-based method, which operates in an open-loop manner, appears more stable. On the other hand, closed-loop approaches like MTRNN and SCONE might be less stable during control. The disparity between MTRNN and SCONE is slight. In particular scenarios such as red bean scooping, SCONE faced two instances of failure: one was caused by a spillage of red beans, and the other potentially resulted from an occasional misjudgment, leading to a failure to properly scooping the red bean. It is important to note, however, that such occurrences are infrequent.
>
>
> ---
>
> **Q6: The generalization claim seems unstable because as mentioned above, the train food seems very similar to the test food and it would be nice to have tested food with slightly different shapes and textures.**
>
> **A6:** Thank you for your suggestion. We believe that the proposed experimental settings involve distinct characteristics of food items. Apart from particle size, the bowl's food quantity varies. Moreover, we have tested uniquely shaped foods like penne and fruit candy. Last, as mentioned in response to **Q2**, we have tested the model's performance on black tea (liquid) and pudding (semi-solid) in our new experiments. These experiments empirically demonstrate the generalization of our model to diverse food properties.
>
> ---
>
> **Q7: It would be good to have experiments with multiple different spoon shapes.**
>
> **A7:** Thank you for the suggestion. We agree that having the model generalize to different spoons is an important and challenging problem. However, our framework primarily focuses on food understanding rather than tool understanding. We are keen to explore other components to augment the proposed framework to achieve food scooping under diverse scenarios.

---

> > ### Comment · Reviewer_VGVQ · 2023-08-13
> > **Response to Authors**
> >
> > Thank you for the response and the clarifications to my questions. Also, thanks for running experiments on more diverse food types. The demo videos are great! The point about different spoon shapes is also valid. I'm definitely interested to see if your future work can incorporate tool understanding with food understanding.

---

> > > ### Author Response · Authors · 2023-08-14
> > > **Thank you for the acknowledgement!**
> > >
> > > We extend our gratitude for your valuable reviews and suggestions that contribute to the refinement of our efforts! We are dedicated to ongoing framework expansion, aiming to seamlessly integrate tool understanding!

---

### Official Review · Reviewer_BzGd · 2023-07-12

**Confidence:** 5
**Originality:** Excellent
**Technical Quality:** Excellent
**Clarity Of Presentation:** Excellent
**Impact:** 4

**Recommendation:**

Strong Accept: I recommend accepting the paper and will argue for my recommendation even if other reviewers hold a different opinion.

**Review:**

This is a novel method for solving a challenging task. Scooping and active perception are not new in robotics (as the paper points out), but using active perception to perform something akin to latent system identification for non-rigid media like beans and rice is. While the method is fairly complex, the results clearly show that it outperforms the baseline in almost every case. Furthermore, the results are reported as number of success out of total attempts, which is a very intuitive and easy to understand metric. Overall the methodology proposed by this paper shows a novel and interesting way to incorporate active feedback in a challenging manipulation task.

One thing this paper does really well is thoroughness in experimental analysis. Table 1 shows a comparison of the main method with 7 baselines spanning variations on behavioral cloning, kNN-likes (the template methods), and even an RNN. Furthermore, the methods are all evaluated on a variety of foods with very different properties, from rice to oranges! Finally, the ablation study in Table 2, even though only a small part of the paper, does a good job justifying the existence of the various parts of the model.

Overall, there’s not much to detract from in this paper. Would it be nice if there were more than 10 trials per food X algorithm combination? Yes, but as it stands there are already 800 trials evaluated in the paper, which is approaching the limit of what can be done with human supervision, so it’s not reasonable to expect more trials. However, an interesting follow-up work might investigate using automated data collection that doesn’t require human involvement (e.g., scooping back and forth between 2 containers). It would also be interesting to try this method on semi-solid and liquid foods such as jams, puddings, creams, and even water. But none of this is necessary as this is an excellent paper as is (though authors feel free to try that last idea for the rebuttal, it would really elevate the paper quite a bit).


**Quality Of The Limitations Section:**

Limitations are addressed clearly

**Questions For Rebuttal:**

As mentioned above, the paper is in excellent shape as is. Evaluating the policy on other kinds of food (cream, jam, etc.) just to see what it does would add a nice touch to the results, but isn't strictly necessary.

A small nit: On line 170 z_interact is in R^{7 x D} space. Should the 7 be K? If not, can you clarify where that number comes from?

**Robotics Focus:**

Sufficient demonstration on hardware

**Summary Of Paper:**

This paper presents a method for scooping various granular food products using a robot arm and a large spoon.  The robot first interacts with the food by stirring it with the spoon. Then the robot attempts to scoop the food and finally dump it in another container. The observations from both the interact and scoop phases are encoded into 4 latent vectors, which are then fed to a scooping policy. The results show that the robot is able to achieve a 71% success rate on average over all the tasks, and is even able to generalize to food with physical properties slightly outside the training distribution.


**Summary Of Recommendation:**

I recommend acceptance of this paper based on the novelty of its methodology, the challenging nature of the task, and the thoroughness of the results.

---

> ### Author Response · Authors · 2023-08-13
> **Response to Reviewer BzGd**
>
> We thank the reviewer's valuable and helpful feedback. We are pleased that you recognize and appreciate our work! To further demonstrate the capabilities and potential of our framework, we conducted new experiments involving continuous scooping and scooping of liquid/semi-solid foods. The demo videos showcase the arm's ability to successfully perform continuous scooping under complex food conditions, including special foods like black tea and pudding. Please refer to the general response section for more details. We look forward to exploring further possibilities based on the proposed framework for robot food manipulation.
>
> ---
> **Q1: An interesting follow-up work might investigate using automated data collection that doesn’t require human involvement (e.g., scooping back and forth between 2 containers).**
>
> **A1:** Thank you for the suggestion! Indeed, it is an interesting and worthwhile direction for follow-up work. Real-world data collection often requires significant human involvement, consuming considerable time. Establishing an automated data collection method could ease this load. We might explore combining the proposed framework and fast online adaptation strategy such as [1] to automate data collection.
>
> [1] Kuma et al., RMA: Rapid Motor Adaptation for Legged Robots, RSS 2021.
>
> ---
>
> **Q2: Experimenting on semi-solid and liquid foods such as jams, puddings, creams, and even water.**
>
> **A2:** In the new experiments, we tested scooping black tea and pudding, demonstrating our method's generalization ability to diverse food properties. **New demo video can be found [here](https://sites.google.com/view/corlscone/%E9%A6%96%E9%A0%81)**.
>
> ---
>
> **Q3: On line 170 z_interact is in R^{7 x D} space. Should the 7 be K?**
>
> **A3:** Yes, the value 7 represents the parameter K, which indicates the number of frames sampled from the original interacting sequence.

---

> > ### Comment · Reviewer_BzGd · 2023-08-14
> > **Excellent!**
> >
> > I appreciate the authors's response to my comments, in particular the added experimental results. I'm really impressed the method worked so well on liquid (black tea)! That was not expected, but I think it bolsters the claim in the paper that this is a novel and robust methodology.  I've watched the video and it looks good. However, I can't seem to locate a revised manuscript with these changes. Usually that's attached to the rebuttal. Do the authors have a revised version of their paper with these changes that I can review?

---

### Official Review · Reviewer_vCX4 · 2023-07-17

**Confidence:** 4
**Originality:** Good
**Technical Quality:** Good
**Clarity Of Presentation:** Very Good
**Impact:** 3

**Recommendation:**

Weak Accept: I recommend accepting the paper, but will not argue for my recommendation if the majority of other reviewers have a different opinion.

**Review:**

Quality: The paper is technically sound and claims well supported by real-world experiments. The idea of active perception for robot scooping is intuitive and effective.

Clarity: The paper is clearly written and well-organized.

Originality: To the best of my knowledge, this work seems to be a novel combination of methods in robot scooping.

Significance: The idea of active perception has the potential to be applied to other tasks in deformable manipulation since the interaction is a natural way to learn the dynamics of the target objects.

Strengths:
1. The authors show that their method can generalize to an extended set of unseen materials in different settings and outperform baselines.
1. The visualizations of interactive embeddings and attention scores are very compelling.

Weaknesses:
1. I'm a little unsure about how difficult this task is since I imagine an open-loop policy will also have relatively reasonable performance. For example, if the robot inserts the ladle into the bowl, rotates around the $y$-axis by a fixed angle, pushes down, rotates back to align with the horizontal plane, and lifts, how will that end up in the task of scooping? The simplified action space (3D position and the y-axis pitch rotation) also gives the policy learning module some important priors to leverage.
1. The authors show material-wise generalization, but how about generalizing to different spatial locations of the bowls (e.g., what if the target bow is further away), different sizes/shapes/colors of the bowls and ladles?
1. It would make this submission much stronger if the authors could compare against methods beyond the paradigm of behavior cloning and active perception, such as model-based methods [[1](https://arxiv.org/abs/2306.16700)] or more up-to-date imitation learning methods [[2](https://arxiv.org/pdf/1903.01973.pdf)].
1. As the authors mentioned in the limitation section, the hard-coded actions for active perception may limit the possibilities the robot explores and inject strong human priors into the model.

**Quality Of The Limitations Section:**

Limitations are addressed clearly

**Questions For Rebuttal:**

1. Have you ablated different architecture designs for each individual encoder?
1. How would you expect this method to generalize beyond the task of scooping and transferring?

**Robotics Focus:**

Sufficient demonstration on hardware

**Summary Of Paper:**

The paper presents SCONE, a food-scooping robot learning framework that leverages active perception to learn meaningful representations of food. SCONE uses interactive encoding and state retrieval modules to capture essential food characteristics and states. The system exhibits success in real-world experiments, achieving a 71% success rate on unseen food items across varying difficulty levels, outperforming other baselines.

**Summary Of Recommendation:**

The paper presents an effective approach to robot scooping with active perception. It's technically sound and well-supported by experiments. However, the task's difficulty is unclear. Generalization to varying spatial locations, sizes, shapes, and colors of bowls and ladles isn't addressed. Comparisons beyond behavior cloning and active perception, such as model-based or advanced imitation learning methods, would strengthen the paper. Also, the reliance on hard-coded actions for active perception might limit the robot's exploration and inject human priors. Therefore, I think more works are necessary to validate this method's robustness and applicability.

---

> ### Author Response · Authors · 2023-08-13
> **Response to Reviewer vCX4**
>
> We thank the reviewer's valuable and helpful feedback. We have conducted two additional experiments to demonstrate our framework's capability. We tested our model's ability to scoop liquids and semi-solids, yielding promising results. Also, we demonstrate that despite varying initial states, the proposed method can discern the current state of food and achieve continuous scooping success. Detailed experimental descriptions are provided in the general response.
>
> ---
>
> **Q1: Difficulty of this task is limited. Could we apply open-loop policy to achieve relatively reasonable performance?**
>
> **A1:**  We recognize the potential of well-designed open-loop actions to improve the performance of certain scooping tasks. However, our scooping task not only involves successfully scooping up the food but also requires avoiding pushing, spilling, and damaging the food (mentioned in the L121-124 of the main paper). This requires a good understanding of food properties. Therefore, these added definitions of task success contribute to the increased complexity of the overall task.
>
> Regarding open-loop control, we implement the template method (KNN-likes, discussed in L217-220 of the main paper). As reported in our experimental results (Table 1 of the main paper and the new "continuous scooping" task), this method struggles to adapt to environmental changes and adjust actions accordingly. The open-loop method may achieve favorable results in an environment similar to the training data. However, it's challenging to equip the open-loop framework with such adaptability in environments with greater variations.
>
> ---
>
> **Q2: Can the framework generalize to different spatial locations of the bowls, or different sizes/shapes/colors of the bowls and ladles?**
>
> **A2:** We acknowledge the importance of considering different configurations, including spatial locations, sizes, shapes, and colors of containers and spoons. However, as pointed out in the general response, our primary focus remains on food property-wise generalization. If the capabilities of these configurations are required, we can incorporate perception components, such as object detection and pose estimation [1, 2] and implicit perception [3], to identify target locations along with the proposed framework to handle different configurations.
>
> [1] Xiang et al., “PoseCNN: A convolutional neural network for 6D object pose estimation in cluttered scenes” (RSS 2018)
>
> [2] Park et al., “Pix2Pose: Pixel-Wise Coordinate Regression of Objects for 6D Pose Estimation” (ICCV 2019)
>
> [3] Zeng et al., "Transporter Networks: Rearranging the Visual World for Robotic Manipulation" (CoRL 2020)
>
> ---
>
> **Q3: Compare against methods beyond the paradigm of BC and active perception**
>
> **A3:** Thank you for the valuable suggestions. Our setting involves interacting with food items to obtain latent food representations, so we explore different strategies for utilizing interaction data for downstream scooping tasks. We compare our framework with various methods (see Table 1 of the main paper). Through our extensive evaluations, we empirically demonstrate the superiority of the proposed framework.
>
> ---
>
> **Q4: The limitation of using hard-coded actions.**
>
> **A4:** Using hard-coded actions for active perception is a limitation. However, considering the fragility or unique properties of food items, we believe this pre-defined task-specific interaction approach is a sensible choice to avoid damaging foods or address other safety concerns. That being said, we believe the reviewer's question is a critical direction to pursue.  We hope our work could collectively stimulate the community to tackle this important and challenging direction.
>
> ---
>
> **Q5: Have you ablated different architecture designs for each individual encoder?**
>
> **A5:** Yes, we have ablated different architectural designs.
> - For the global-temporal encoder, we tested different input sizes and CNN architectural designs (e.g., number of layers). We found that the current design works favorably.
> - Regarding the interactive encoder, we tested CNN with triplet loss, sequential encoding with LSTM, and dynamics modeling with VAE. However, the results remained the same or even deteriorated compared to the current CNN architecture. We believe this could be due to our small dataset, which makes training complex architectures more challenging.
> - For the state retrieval module, we chose the attention-based architecture to implement our idea. Therefore, our testing focused on cropped observation and feature dimension variations to determine the most suitable parameters.

---

> > ### Author Response · Authors · 2023-08-13
> > **Response to Reviewer vCX4 (Part 2)**
> >
> > ---
> >
> > **Q6: How would you expect this method to generalize beyond the task of scooping and transferring?**
> >
> > **A6:** Thanks for pointing out the important question! We strongly believe in the proposed framework being applied to various tool use for food manipulation. One might need to define appropriate actions in the active perception stage to enable these downstream tasks. For instance, if cutting is the downstream task, we can define pre-cutting actions to apply certain force on foods. We could acquire representations of food items for cutting through the interactive process. We are actively working toward applying the framework to other tasks and hope to share our findings with the community.

---

> > > ### Comment · Reviewer_vCX4 · 2023-08-14
> > > **Response to Authors**
> > >
> > > Thank you for the response and clarifications! I'm very interested to see how the proposed framework works on other tasks.

---

> > > > ### Author Response · Authors · 2023-08-15
> > > > **Thank you again for your comments**
> > > >
> > > > We want to clarify that it is challenging to complete a whole cutting experiment due to the time constraints of the rebuttal period. We will continue to conduct relevant research in this area. If you have any further questions or concerns about our work that we can address, please feel free to let us know. Thank you!

---

### Official Review · Reviewer_S1uK · 2023-07-19

**Confidence:** 5
**Originality:** Fair
**Technical Quality:** Fair
**Clarity Of Presentation:** Good
**Impact:** 3

**Recommendation:**

Weak Accept: I recommend accepting the paper, but will not argue for my recommendation if the majority of other reviewers have a different opinion.

**Review:**

Writing Quality: The writing quality was pretty good. There were only a few mistakes that I noticed in the supplementary video and some places in the paper.

Clarity: The paper was pretty clear, although some of the figures could be fixed. I think Figure 2 could be improved as it initially confused me with only arrows going into the state retrieval module and nowhere else.

Originality and Significance: This paper seems to be a combination of different previous ideas applied to the task of scooping. One would be active perception, another is an attention module, and the last is a concatenation of previous frames into an encoder. I would not consider this a very novel paper.

Strengths: The authors did do a lot of real world scooping experiments. Their method does seem to improve on their other baselines and ablations.

Weaknesses: I have a lot of concerns for this paper. I will put most of this section in the questions for Rebuttal.

**Quality Of The Limitations Section:**

Limitations are not well addressed

**Questions For Rebuttal:**

1. The limitations section only mentioned the interaction action component, but I think there are more limitations.
2. The authors didn't use any proprioceptive feedback (force torque sensing) from the robot, which could provide many benefits during the active perception stage. Many previous papers use proprioceptive feedback during active perception.
3. The self defined criteria for success is not great in my opinion. The authors reset after each trial to a complete starting point with one bowl with the specified amount and the other is empty. Instead, I believe a better metric would be number of scoops before 1 bowl is completely empty with extra penalties for spilling or damaging the food.
4. I believe the task is a bit too simple. Many people have done scooping from bowls in the past and I feel like a well hard coded scoop action could get a high score on the red beans by just shaking extra beans off in the bowl before transport. Instead I think you should vary the locations of bowls and show that the closed loop control of the robot can still scoop the items from any location using your pipeline.
5. There is a lack of citations. Some recent papers would be: (a) Excavation learning for rigid objects in clutter by Qingkai Lu and Liangjun Zhang, (b) Soil-adaptive excavation using reinforcement learning by Pascal Egli, Dominique Gaschen, Simon Kerscher, and Dominic Jud, and Marco Hutter, (c) Uncertainty-Aware Self-Supervised Target-Mass Grasping of Granular Foods by Kuniyuki Takahashi, Wilson Ko, Avinash Ummadisingu, Shin-ichi Maeda, (d) Learning robotic manipulation of granular media by Connor Schenck, Jonathan Tompson, Dieter Fox, Sergey Levine, and (e) Learning audio feedback for estimating amount and flow of granular material by Samuel Clarke, Travers Rhodes, Christopher G. Atkeson, and Oliver Kroemer
6. There should be different container sizes and shapes like plates and larger bowls. But if it requires collecting a massive amount of data to generalize to different containers, then I think there may be an issue with the approach.
7. There was not any water / liquid substances like jello or pudding, which I think is the most challenging. I feel like scooping of granular material like Connor Schenck's paper has already shown promise if given enough data. Sticky or liquid scooping is far more difficult with higher stakes. The insights from a paper on that would be far more impressive and novel.

**Robotics Focus:**

Sufficient demonstration on hardware

**Summary Of Paper:**

The authors of this paper explored the problem of scooping various unseen food items using a Franka with a ladle. They used active perception with some stirring actions and recorded 7 frames using a pre-specified hard coded motion. Then they ran the 7 RGBD frames into an encoder to get a latent embedding for the food item. Next, they used a closed loop policy that takes in the embedding from the active perception, 10 previous RGBD frames, and a state retrieval module that uses cropped rgb images and attention as input into a policy that outputs a delta action. Their action space is the robot's XYZ and y-axis pitch rotation of the end-effector. They compared their method with a variety of other behavior cloning approaches and templated approaches. They reported a 70% success rate using their metric of no spilling, filling up at least 1/3 of the spoon, and not damaging the food items. They also tested on black tea and pudding, which showed some fair results.

**Summary Of Recommendation:**

I updated my review to a weak accept because the authors did test on new liquid/viscous objects and the performance was still acceptable. In addition, they added continuous scooping results. While I still believe a well formulated baseline using some image segmentation and a good hard coded scooping action could achieve similar results. It does seem good that it works in a closed loop manner. I would still prefer if there were different containers and tools as well as the skill adapting when things change in the environment such as human interference. I will not argue for acceptance, but it has improved from its original submission.

---

> ### Author Response · Authors · 2023-08-13
> **Response to Reviewer S1uK**
>
> We thank the reviewer for valuable and helpful feedback.
>
> ---
> **Q1: The limitations section only mentioned the interaction action component, but I think there are more limitations.**
>
> **A1:** We greatly value the reviewer's insightful suggestions. In response to your questions, we conducted new experiments to verify the effectiveness of the proposed framework. For more details, please refer to the following responses. We hope that these new experiments and answers can address your concerns.
>
> ---
> **Q2: The authors didn't use proprioceptive feedback (force torque sensing) from the robot, which could benefit active perception.**
>
> **A2:** While acknowledging the potential benefits of proprioceptive feedback in enhancing performance, our experimental results already showcase the effectiveness of our current setup in learning food item properties (Figure 3 of the main paper) through active perception, aligning with our primary goal, and the framework's performance outperforms other baselines (Table 1 of the main paper). In our future work, We will explore the potential of incorporating proprioceptive feedback to capture food properties better and improve scooping.
>
> ---
> **Q3: Task is a bit too simple. Well hard coded scoop action could get a high score on the red beans by shaking extra beans off before transport. Instead I think you should vary the locations of bowls and show that the closed loop control of the robot can still scoop the items from any location.**
>
> **A3:** Thank you for your insightful question. We respectfully agree with the reviewers that it is possible to design hard-coded algorithms to achieve favorable performance for scooping. However, our primary objective is to demonstrate the feasibility of the interact-then-manipulate framework to acquire food properties. Therefore, our task design primarily emphasizes enabling the model to adopt appropriate policies based on food properties and challenging cases involving uniquely shaped and textured foods. As mentioned in your question, we believe the goal can be achieved by integrating other existing perception components with the proposed framework to enable the applicability to containers placed at arbitrary positions. Specifically, we can explore explicit perception, such as object detection and pose estimation [1, 2], or implicit perception [3] to identify target locations.
>
> [1] Xiang et al., “PoseCNN: A convolutional neural network for 6D object pose estimation in cluttered scenes,” RSS 2018.
>
> [2] Park et al., “Pix2Pose: Pixel-Wise Coordinate Regression of Objects for 6D Pose Estimation,” ICCV 2019.
>
> [3] Zeng et al., "Transporter Networks: Rearranging the Visual World for Robotic Manipulation," CoRL 2020.
>
> ---
>
> **Q4: There should be different container sizes and shapes like plates and larger bowls. But if it requires collecting a massive amount of data to generalize to different containers, then I think there may be an issue with the approach.**
>
> **A4:** We sincerely agree that generalizing different container sizes and shapes is essential in real-world applications. However, this is not the primary objective of this work. Nevertheless, we are keen to explore the potential of the proposed framework for the suggested settings by combining it with existing perception components, as highlighted in **General Response** and **Q3**. With these modules, we do not need to collect a massive amount of data for different containers.
>
> ---
> **Q5: There was not any water / liquid substances like jello or pudding, which I think is the most challenging.**
>
> **A5:** Thank you for the great suggestions! Indeed, scooping liquid/semi-solid food successfully is a significant challenge. The general response shows that the proposed framework successfully demonstrates the capability to scoop black tea and pudding.  **NEW Demo Videos can be found [here](https://sites.google.com/view/corlscone/%E9%A6%96%E9%A0%81)**.

---

> > ### Author Response · Authors · 2023-08-13
> > **Response to Reviewer S1uK (Part 2)**
> >
> > **Q6: The self defined criteria for success is not great in my opinion. The authors reset after each trial to a complete starting point with one bowl with the specified amount and the other is empty. Instead, I believe a better metric would be number of scoops before 1 bowl is completely empty with extra penalties for spilling or damaging the food.**
> >
> > **A6:** Thank you for your suggestions. We believe your suggestion is a valuable assessment approach. From our perspective, the primary focus of addressing the scooping task revolves around "how to take corresponding actions based on the properties of the food." While analyzing the model's ability from the aspect of emptying the bowl could be an evaluation protocol of the proposed method, it focuses on developing algorithms to achieve optimal efficiency in scooping instead of thoroughly evaluating the model's understanding of food properties. In the rebuttal, we have additionally introduced a continuous scooping experiment. For detailed information, please refer to the second point in the general response section. In the experiment, the models need to accurately acquire the characteristics and states of the food to achieve continuous scooping. This experiment offers another perspective to evaluate the model's abilities.
> >
> > ---
> > **Q7: Lack of citations.**
> >
> > **A7:** Thank you for pointing out these valuable references. While these papers are related to scooping tasks or granular food manipulation, we believe the key distinction between them and our work is that we focus on "adapting different scooping strategies according to food properties and states." The methods and insights presented in the above-mentioned papers can be applied to our scooping task to some extent. Still, further refinements are necessary due to the differences in task objectives.  Regarding methodology design, we can draw inspiration from their mechanisms to enhance the task success rate. We will include the two papers [4, 5] in the Robotic Food Scooping section.
> >
> > [4] Schenck et al., “Learning Robotic Manipulation of Granular Media,” CoRL 2017.
> >
> > [5] Clarke et al., "Learning Audio Feedback for Estimating Amount and Flow of Granular Material," 2018

---

> ### Comment · Reviewer_S1uK · 2023-08-14
> **Response to Rebuttal**
>
> I am glad that the authors tried new items like the pudding and black tea. I also watched the continuous scooping gifs. They seemed fine and it is true, it would be an optimization problem to scoop the entire amount of items as fast as possible instead of just scooping success. Although I still believe that a hard coded motion with some initial info about the locations of the items would be able to perform similarly, I will raise my evaluation from a weak reject to a weak accept. It would have been better to show scooping on more items where the scooping forces would need to be changed like ice cream when it is hard or melting or maybe some more viscous sauces that some chefs pick up before stir frying.

---

> > ### Author Response · Authors · 2023-08-15
> > **Thank you again for your response!**
> >
> > We agree that certain scenarios are well-suited for employing hard-coded motions in the food scooping task. However, in specific scenarios, closed-loop policies also demonstrate significant potential. Perhaps combining the strengths of these two methods could contribute to the entire community's benefit.
> >
> > Furthermore, we intend to investigate unique food scooping scenarios, such as dealing with ice cream before/after melting or handling high-viscosity sauces, as part of our future research. We believe that incorporating proprioceptive feedback could contribute to enhancing the task success rate.
> >
> > If you have any further questions or concerns about our work that we can address, please feel free to let us know. Thank you!

---

### Author Response · Authors · 2023-08-13
**General Response #1**

We thank valuable and insightful suggestions from all the reviewers. We are delighted that reviewers recognize our work has several strengths, including:
* **Task setting/Problem formulation/Significance**: a challenging manipulation task (BzGd), one of the few to introduce closed-loop control during food manipulation (VGVQ), and the idea of active perception has the potential to be applied to other tasks in deformable manipulation (vCX4).
* **Method**: The paper is technically sound / The idea of active perception for robot scooping is intuitive and effective / a novel combination of methods in robot scooping (vCX4). Novel perspectives on implicit food representation through active perception, along with closed-loop food manipulation, stand as unique contributions (BzGd and VGVQ).
* **Experiment**: Numerous real-world scooping experiments showed the proposed method outpeform other baselines. (S1uK, vCX4, and BzGd)
* **Metric**: very intuitive and easy to understand metric (BzGd)
* **Explainability of the proposed method**: The visualization enhances the explainability behind policy learning (vCX4).
* **Paper Quality**: High writing quality (S1uK, vCX4, BzGd, and VGVQ), high originality (vCX4, BzGd, and VGVQ), high technical quality (vCX4 and BzGd)

---

**Summary of our work:**

Scooping diverse food properties, including deformability, fragility, fluidity, or granularity, remains challenging for existing representations of food items. In this work, we propose a novel "interact-then-manipulate" framework involving an "Interact" stage to acquire implicit representations of food items and a "Manipulation" stage to integrate the representations for closed-loop scooping policy. We conduct extensive experiments to prove the proposed framework can succeed at scooping unseen properties of food items and outperforming the baselines. We hope to stimulate the community to explore the framework's potential for other food manipulation tasks, such as cutting and skewering, where diverse food properties involve.

---

> ### Author Response · Authors · 2023-08-13
> **General Response #2**
>
> We found that reviewers were most concerned about the **limited complexity/difficulty of the task**. There are several aspects to address regarding the complexity of the proposed task:
>
> **1. Generalization to other configurations**: We acknowledge the importance of considering different configurations of generalization, including spatial locations, sizes, shapes, and colors of containers and spoons for robotic scooping. However, our primary focus remains on food property-wise generalization. The proposed "interact-then-manipulation" framework can better address the issue of food property-wise generalization. We believe the goal can be achieved by integrating other existing perception components, such as object detection and pose estimation, with the proposed framework to enable the applicability to these configurations.
>
> **2. Generalization to diverse food properties:** To strengthen the effectiveness of the proposed framework for diverse food properties, we conduct two additional experiments: (1) liquid/semi-solid food scooping and (2) continuous scooping. **NEW Demo Videos can be found [HERE](https://sites.google.com/view/corlscone/%E9%A6%96%E9%A0%81)**. The details of the tasks are discussed in the following.
>
> * **Liquid/semi-solid food scooping**
> Scooping liquid/semi-solid food successfully is a significant challenge. We design a new experiment to apply the proposed method to scoop black tea and pudding. This experiment effectively demonstrates the framework's ability to generalize to unseen food items, as our training data did not include any liquids or semi-solid food items. The environment setting and the evaluation metrics are the same as the experiments detailed in the paper. The experimental results are shown as follows:
>
>    |           | Black Tea | Pudding |
>    | --------- |:---------:|:-------:|
>    | **SCONE** |   8/10    |  5/10   |
>
>
>     From the results, it can be seen that the proposed method can generalize to unseen foods and achieve a success rate of over 50%. It shows the potential of our approach to be applicable across diverse food characteristics.
>
> * **Continuous scooping**
> The model repetitively transfers food from one bowl to another until it fails (scoops up nothing). A notable change from the original experiment is removing the criterion involving scooping less than one-third of a spoon as a failure. In the experiment, the models need to accurately acquire the characteristics and states of the food to achieve continuous scooping under varying initial conditions. We compared three methods in scooping tasks with red beans and penne pasta. The table below displays the average number of successful continuous scoops over five trials for each food:
>
>
>    **AS:**       average number of successful scoops per trial,
>    **Max/Min:**  the maximum and minimum number of scoops in a single trial,
>    **Penalty:**  the total instances of food spillage and food/setting damage, divided by the total number of scoops.
>    |                         | Penne |         |         | Red Bean |         |         |
>    |:-----------------------:|:-----:|:-------:|:-------:|:--------:|:-------:|:-------:|
>    |                         |  AS   | Max/Min | Penalty |    AS    | Max/Min | Penalty |
>    | **Classified Template** |   0   |   0/0   |   0%    |   5.4    |   8/2   |   29%   |
>    |        **MTRNN**        |  2.8  |   4/2   |   42%   |   2.8    |   4/2   |   21%   |
>    |        **SCONE**        |  3.2  |   4/2   |   25%   |   13.4   |  14/5   |   22%   |
>
>     From the results, we observed that our method performs better than other baselines, whether in the simple red beans or the more complex penne scooping task. Our framework showcases the ability to successfully scoop food across intricate variations in food states, with a lower proportion of food/setting spillage and damage.
>
>
> **3. How well-designed hard-coded actions work:**  We respectfully agree with the reviewers that well-designed hard-coded algorithms can achieve favorable performance in scooping tasks. However, our food scooping task involves successfully scooping up the items and also requires avoiding pushing, spilling, and damaging the food (mentioned in the L121-124 of the main paper). When dealing with diverse properties and states of foods, these open-loop methods struggle to adapt to environmental changes. If we aim to equip the open-loop framework with such adaptability, it requires careful designs and hyperparameter tuning, yet it remains challenging to address all possible scenarios.

---

### Decision · Program_Chairs · 2023-08-30

**Decision:**

Accept (Poster)

**Comment:**

This submission first received mixed reviews.  The authors did a good job during the rebuttal phase, and all reviewers recommended acceptance after discussion.  The AC agrees with the recommendation.  Some reviewers remain concerned about the generalization of the proposed method.  The authors are encouraged to further explore this direction based on the reviews.